# COmpliance with pandemic COmmands Scale (COCOS): The relationship between compliance with COVID-19 measures and sociodemographic and attitudinal variables

**Fabia Morales-Vives**[1,2☯]*, **Jorge-Manuel Dueñas**[1,2☯], **Pere J. Ferrando**[1,2‡], **Andreu Vigil-Colet**[1,2‡], **Maria Dolores Varea**[3‡]

**1** Psychology Department, Universitat Rovira i Virgili, Tarragona, Spain, **2** Research Center for Behavior Assessment (CRAMC), Tarragona, Spain, **3** Pedagogy Department, Universitat Rovira i Virgili, Tarragona, Spain

☯ These authors contributed equally to this work.
‡ These authors also contributed equally to this work.
* fabia.morales@urv.cat

**Data Availability Statement:** All data are within the Zenodo repository: DOI 10.5281/zenodo.5828250.

## Abstract

Several studies in different countries have reported that part of the population does not fully comply with the measures recommended to prevent COVID-19, and therefore poses a risk to public health. For this reason, several measures have been developed to assess the level of compliance, although many of them have methodological limitations or do not include a comprehensive set of items. The main goal of the current study was to develop a new instrument with suitable psychometric properties, which includes a more complete set of items and controls the impact of acquiescence bias. The participants were 1410 individuals (59.2% women) from Spain, who answered the new questionnaire and several items on sociodemographic and attitudinal issues. Exploratory and confirmatory factor analyses were carried out, and the results suggested that only one content factor was underlying the data. This solution was replicated in a different subsample, which shows the stability of the solution. Furthermore, the relationships between the scores of the new questionnaire and the sociodemographic and attitudinal variables are similar to those obtained in previous studies, which can be regarded as evidence of the validity of the new questionnaire.

## Introduction

The new coronavirus SARS-CoV-2 was first detected on December 1, 2019, in Wuhan (People's Republic of China) and, since then, it has spread throughout the world, causing thousands of deaths and hospital admissions. For many months there were no treatments or vaccines against the disease caused by this new coronavirus, called COVID-19, so the first contingency measures were designed to regulate the behavior of the general public, with measures and restrictions such as lockdown, social distancing, the use of masks, handwashing, etc.

**Funding:** This research was supported by a grant from the Spanish Ministry of Economy and Competitivity (PID2020-112894GB-I00) and a grant from the Catalan Ministry of Universities, Research and the Information Society (2017 SGR 97). The funders had no role in study design, data collection and analysis, decision to publish, or preparation of the manuscript.

**Competing interests:** The authors have declared that no competing interests exist.

Although several vaccines against COVID-19 are now available, these preventive measures are still necessary to prevent the disease from spreading and to protect the most vulnerable citizens. Moreover, some experts consider that COVID-19 may become endemic, and that the virus may mutate and infect vaccinated people, so some of the preventive measure may still be necessary for a long time to come. However, various governments are facing the challenge of ensuring that their citizens comply with the preventive measures as the pandemic evolves.

Several studies in different countries have reported that part of the population does not fully comply with the measures designed to prevent COVID-19 [1–3], and so pose a risk to public health. The study by Park et al. [4] shows that in the United States the measures with the highest levels of non-compliance are hand washing and social distancing. On the other hand, the study by Siebenhaar et al. [5] shows that in Germany the measure with the highest level of non-compliance is wearing the face mask.

Various sociodemographic characteristics have been related to non-compliance with the restrictions. Of these, sex is one of the most reported variables, and many studies have found that women are more likely to comply with the restrictions than men [1,2,4]. Age also appears to be another relevant variable, and, according to many reports, young people are more likely to ignore the restrictions and take fewer prevention measures than older people [4,6]. The fact that people over 65 are especially vulnerable to this disease, with a higher risk of suffering serious consequences and even death than younger people [7] may indeed partly explain this result. It seems, also, that married people and people with children are more likely to follow the preventive measures, possibly because they are more worried about their families [2]. However, income level does not seem to be a relevant variable for predicting adherence to guidelines [2,8].

As far as attitudes and beliefs are concerned, people who trust the health system [9], the government [1,2], and science [10] are more likely to comply with COVID-19 prevention guidelines. In addition to distrust in government and experts, receiving mixed messages about the pandemic may also lead to lower levels of compliance [11].

The study by Monzani et al. [12] showed that optimistic people are more likely to develop an optimistic bias about COVID-19, and judge their risk of getting infected to be lower than that of others. Along the same lines, a study carried out in the Netherlands, Germany, Greece, and the USA showed that people with an optimistic bias were less likely to engage in protective behaviors such as hand washing or social distancing [13]. So, optimism and the optimistic bias about the COVID-19 may play a role in some decisions about compliance with the preventive measures.

Several instruments have been developed to assess the level of compliance, but many of them have methodological limitations. For example, Siebenhaar et al. [5] developed the Compliance Index, which has 13 items on behaviors such as staying at home, following recommended hygiene regulations, social distancing, wearing a face mask, going to the gym, going to a party, going to a restaurant, taking a trip, visiting family, etc. The dimensionality and structure were assessed using principal-component analysis with varimax rotation, which retained a bidimensional solution. The first component included those items related to leisure and family activities (parties, restaurants, visiting family, etc.), whereas the second included the remaining recommendations (hygiene, face mask, keeping distance, public transport, etc.). The strength of this second component and the reliability of the scores derived from it were rather low (Cronbach's $\alpha = 0.44$), and a careful item analysis was not undertaken. Rather, the authors decided not to remove any items because, by fiat, all behaviors were considered to be highly relevant to the prevention of COVID-19. Lep et al. [14] also developed a 9-item instrument for assessing compliance, and decided that three common factors were underlying the data: personal hygiene, social contacts, and preparatory behaviors. However, the first and third factors were only defined by two items each, which means that (a) the strength and construct validity of the solution are rather questionable, (b) the factors may well be artifactual, and

defined by an item doublet sharing specific content, and (c) scores derived from this solution will necessarily not be reliable for diagnostic purposes [15]. Indeed, the estimated reliabilities of the scores derived from this solution were very low for these two factors ($\rho_{\theta\theta}$ = .55 and $\rho_{\theta\theta}$ = .47, respectively). Triberti et al. [16] used a 16-item set based on the COVID-19 guidelines by the Italian Ministry of Health, and their Exploratory factor analyses revealed two factors (Healthy behaviors and Bad behaviors) although, again, the scores derived from one of these factors were quite unreliable (Cronbach's α = 0.50). Galasso et al. [17] also developed a compliance index, which includes behaviors such as washing hands more often, coughing into one's elbow, not greeting people by shaking hands or hugging, avoiding crowded places, keeping your distance from others, staying at home, not visiting friends. However, no dimensionality or structural analyses were performed on the data, and no evidence of the psychometric functioning of the measure was provided. These criticisms can be extended to several studies that used similar item sets [1,18,19]. Finally, at the item-content level, one limitation of many previous studies is that they include only a few preventive measures [20–22], particularly the use of a face mask, social distancing and staying at home, which does not sufficiently reflect the heterogeneity of measures proposed during the pandemic (i.e. the domain of measurement is not fully sampled).

While the review above does not claim to be exhaustive, we believe that the shortcomings noted above warrant new psychometrical developments for assessing compliance with COVID-19 preventive measures. So, the main goal of the current study is to develop a new instrument, with suitable psychometric properties, which include a more complete set of items sampling the existing range of preventive measures. We also aim for the questionnaire to control the impact of acquiescence bias so that the factor structure is the best possible [23]. A second goal is to determine the extent to which the Spanish people comply with these measures or not. Furthermore, as previous studies show that there are sex differences in the compliance of preventive measures [1,2,4], we aim to determine if these sex differences are also found in the Spanish population, and whether there are also sex differences in the perception of the usefulness of these measures. In fact, we expect to find that women have higher levels of compliance and higher levels of trust in the usefulness of the preventive measures. Finally, in terms of validity evidence based on the scores of our instrument, a fourth goal is to determine whether the relationships between compliance and other variables are consistent with those obtained in previous studies. More specifically, we expect to find higher levels of compliance in older people, and lower levels of compliance in optimistic people. However, we did not expect that household income is a relevant variable for predicting adherence to guidelines, as previous studies suggest [2,8]. We also expect to find higher levels of compliance in people afraid of being infected or afraid a loved one might become infected, people concerned about the health crisis, people who trust in experts and in the health system, and people who suffer a chronic disease. We also expect to find that people who have received at least one dose of a vaccine will have lower levels of compliance because of the sense of security that the vaccine may provide. Furthermore, we expect to find higher levels of compliance in people whose main source of information is official channels and people who trust in the usefulness of the preventive measures. Finally, we expect that people with children, married people, and people with higher educational levels tend to respect preventive measures to a greater extent.

## Materials and methods

### Participants

The participants were 1410 individuals (59.2% women) who were resident in Spain. They were between 15 and 76 years old (M = 30.9, S.D. = 15.7). More specifically, 28% were adolescents

(between 15 and 17 years old), 29% were young people (between 18 and 29 years old) and 43% were 30 years old or older. Only 8.3% of participants had received at least one dose of a vaccine against COVID-19.

In terms of academic level, 0.3% had not finished primary education, 3.1% had finished primary education, 40.5% had secondary education, 36.1% had finished a degree and 20% had postgraduate studies. The participants described themselves as students (49.6%), employees (43.6%), unemployed (1.8%), in temporary layoff (0.8%), or in other situations (4.2%).

The household income was lower than 1000 euros in 7.0% of the sample, between 1000 and 2499 in 44.5% of the sample, between 2500 and 3999 in 31.1%, between 4000 and 5499 in 12.0%, and higher than 5500 in 5.3%.

A total of 60.4% of the sample were single, 24.8% were married, 3.5% were divorced or separated, 10.3% lived with their partner without being married, and 1.0% were widowed. A total of 70.2% of participants did not have children.

## Measures

**Sociodemographic variables.**   The sociodemographic variables were sex and age. Adults were also asked for their marital status (1. Single, 2. Married, 3. Living with a partner but not married, 4. Divorced or separated, and 5. Widowed), if they had children (1 = Yes, 0 = No) and the household income per month (1. Less than 1000 euros, 2. Between 1000 and 2499 euros, 3. Between 2500 and 3999 euros, 4. Between 4000 and 5499 euros, and 5. More than 5500 euros).

**Compliance with preventive measures.**   Participants answered a questionnaire measuring their compliance with the rules governing the COVID-19 pandemic. The questionnaire, entitled COmpliance with pandemic COmmands Scale (COCOS), was specially developed for this study. It consists of 27 Likert-type items with five response options (1 = completely disagree to 5 = completely agree).

**Feelings and opinions about the pandemic.**   Participants answered the following five binary items (1 = Yes, 0 = No) focused on their feelings and fear of the pandemic, and their trust in experts and the health system: Are you worried about the health crisis caused by COVID-19?; Are you afraid of being infected with COVID-19?; Are you afraid that a loved one might be infected with COVID-19?; Do you trust what the experts (epidemiologists, doctors, etc.) have to say about COVID-19?; Do you trust the health system?

**Questions about health and vaccination.**   Participants were asked if the suffered any chronic disease and if they had been infected with COVID-19 at some point. If their answer to the last question was "Yes", they were further asked if they had suffered severe symptoms. Participants were also asked if they had been vaccinated against COVID-19. If their answer was "No", they were further asked if they intended to be vaccinated against COVID-19.

**Optimism.**   One item asked participants to indicate the extent to which they considered themselves to be optimistic, on a scale of 1 to 10 ("*En una escala de 0 a 10, ¿hasta qué punto te consideras una persona optimista?*"). This item assesses general optimism, so it does not refer specifically to COVID-19.

**Opinion about the usefulness of the preventive measures.**   One item asked participants to indicate which of the following measures they considered useful for preventing COVID-19 (respondents could choose more than one option): use of masks in public places, social distancing, hand hygiene, time restrictions (e.g. curfew), mobility restrictions (e.g., forbidden to leave the municipality at weekends).

**Information channels.**   One final item asked participants what information channels they mainly used to get information about COVID 19 (respondents could choose more than one

option): official channels (television, press, government communications, etc.), social networks (WhatsApp, Facebook, Instagram, etc.), websites with alternative information to that offered by the official channels, or others.

## Procedure

This study was approved by the Research and Innovation Ethics Committee (CEIPSA) of Universitat Rovira i Virgili (CEIPSA-2021-PR-0002). We also obtained the written informed consent from all participants.

The questionnaires were administered online, through a survey designed for this purpose, from February to April 2021. The survey included information about how participants should complete the questionnaires. The exclusion criteria for participating in this study were being under 15 years old, not resident in Spain, or not providing informed consent. Participants had to accept the conditions of the study before participating. They were informed that they could drop out at any time. Questionnaires were anonymous, and confidentiality and data protection were guaranteed.

We decided to administer the survey online because the restrictions due to the pandemic made it difficult to travel and administer the questionnaires in person. In order to recruit as heterogeneous a sample as possible, we disseminated the survey in several ways. Some of the participants were recruited through WhatsApp groups, Facebook and Twitter, and we were assisted by our university and several Spanish associations. The project was also announced in the mass media, which provided the link to the questionnaire on their websites. We also contacted high schools from different regions of Spain so that teachers could answer the survey, and disseminate it to their students aged 15 and older. We did not ask for the informed consent of the parents of participants aged 15–18 years old because we could not go in person to the schools to do so, and also because some of the adolescents accessed the survey through the social networks. However, according to Spanish legislation the personal data of minors can be processed with their consent when they are older than 14 (article 7 of *Organic Law 3/2018, of 5 December, on the Protection of Personal Data and Guarantee of Digital Rights*). Once the participants had finished the questionnaire, the website allowed them to share it with other people on the social networks (e.g. WhatsApp and Facebook). This involves a non-probabilistic sampling procedure known as "snowball" [24]. Evidence so far suggests that the online administration of questionnaires provides similar results to those found using paper & pencil administrations [25].

The items of the *COCOS questionnaire* were written by two researchers with experience in developing questionnaires. They first reviewed the items used in previous studies to assess compliance with COVID-19 preventive measures [e.g., 1,5], the measures and recommendations proposed in Spain by the Spanish government [26] and the health departments of the autonomous communities [27], and the international recommendations such as those proposed by the Centers for Disease Control and Prevention [28] and the World Health Organization [29]. The preventive measures recommended by these organizations include the following areas: use of face mask, social distance, hand hygiene, cover the nose and mouth with the crook of the elbow or a handkerchief, compliance with curfews, compliance with mobility restrictions, isolation in case of presenting symptoms or being diagnosed with COVID-19, isolation in case of having had contact with a person diagnosed with COVID-19, and others (avoid parties, crowded places, public transport, shaking hands, travelling without a justified reason, drinking and eating with people who you do not live with, and touching own's eyes, nose and mouth). On the basis of all this information, they wrote a pool of 27 items about compliance with prevention measures for each of these areas, as can be seen in Table 1. Two

**Table 1. Loading matrix obtained from the exploratory factor analysis, means and standard deviations of the whole sample.**

| Item | Loadings | |
|---|---|---|
| | AQ | F1 |
| 1. When I cough or sneeze, I cover my nose and mouth with the crook of my elbow or a handkerchief. *(Cuando toso o estornudo, me cubro la nariz y la boca con el codo flexionado o con un pañuelo)* | .08 | .37 |
| 2. Sometimes I do not wear my face mask properly and it does not cover my nose or my mouth. *(A veces llevo la mascarilla mal puesta, sin cubrir la nariz o la boca)* | .40 | -.58 |
| 3. I avoid crowds. *(Evito las aglomeraciones)* | -.03 | .67 |
| 4. I often touch my eyes, nose and mouth. *(A menudo me toco los ojos, la nariz y la boca.)* | .26 | -.43 |
| 5. I respect lockdowns and try not to leave home. *(Durante los confinamientos, respeto esta medida y evito salir de casa)* | .04 | .64 |
| 6. When I am in a queue, I keep a safe distance from the person in front. *(Cuando estoy haciendo cola, mantengo la distancia de seguridad con la persona que va delante)* | .01 | .67 |
| 7. When I go to bars and restaurants, I drink and eat with people who I do not live with. *Cuando voy a bares y restaurantes, comparto bebida o comida (con personas con las que no convivo)* | .32 | -.53 |
| 8. I try to wear masks that are in good condition. *(Procuro usar mascarillas que estén en buenas condiciones)* | .08 | .63 |
| 9. I avoid coming into contact with the elderly and people who are ill. *(Evito el contacto con personas mayores o con persones que sufren enfermedades crónicas)* | .16 | .46 |
| 10. I use public transport although it is not essential for me to do so. *(Utilizo el transporte público aunque no sea indispensable)* | .34 | -.47 |
| 11. When I go to a bar, I prefer to sit outside than inside. *(Cuando voy a un bar, prefiero estar en la terraza que en el interior)* | .12 | .54 |
| 12. If someone with whom I have been in contact recently were suspected to have COVID-19, I would isolate and avoid contact with others. *(Si una persona con la que he mantenido contacto en los últimos días, fuera sospechosa de sufrir COVID-19, me aislaría y evitaría el contacto con los demás)* | .09 | .72 |
| 13. I often forget to wash my hands when I get home. *(A menudo se me olvida lavarme las manos al llegar a casa)* | .22 | -.46 |
| 14. Whenever I get the opportunity, I travel and go sightseeing. *(Cuando puedo, aprovecho para viajar y hacer turismo)* | .28 | -.44 |
| 15. When I am in a bar or a restaurant, I wear my mask if I am not eating or drinking. *(Cuando estoy en un bar o restaurante, llevo puesta la mascarilla si no estoy comiendo ni bebiendo)* | -.02 | .64 |
| 16. I visit family members with whom I do not live. *(Visito a familiares con los que no convivo)* | .23 | -.41 |
| 17. If I had symptoms (temperature, cough, etc.), I would isolate in my room and avoid contact with other people. *(En caso de presentar síntomas (fiebre, tos, etc.), me aislaría en mi habitación y evitaría el contacto con otras personas)* | .17 | .67 |
| 18. In social situations, I avoid shaking hands. *(En situaciones sociales, evito dar la mano a los demás)* | -.10 | .75 |
| 19. I eat out with friends. *(Voy a cenas o comidas con amigos)* | .37 | -.63 |
| 20. If I am diagnosed with COVID-19, I will try to stop other people from finding out even if I have been in contact with them. *(En el caso de ser diagnosticado/a de COVID-19, intentaré que el resto de personas no se enteren, aunque haya mantenido contacto con ellas)* | .37 | -.41 |
| 21. I respect all the travel restrictions where I live (for example, prohibition on leaving the municipality or region). *(Respeto las restricciones de movilidad impuestas donde vivo (por ejemplo, prohibición de salir del municipio o comarca))* | -.02 | .58 |
| 22. I take off my mask in the street if I have to use my mobile. *(Me quito la mascarilla en la calle para hablar por el teléfono móvil)* | .38 | -.56 |
| 23. I go to parties. *(Voy a fiestas)* | .46 | -.69 |
| 24. If I were to be diagnosed with COVID-19 but had no symptoms, I would carry on as normal. *(En el caso de ser diagnosticado/a de COVID-19 y no presentar ninguna sintomatología, seguiré con mi actividad habitual)* | .38 | -.56 |
| 25. When I am not at home, I try not to spend too much time in enclosed spaces with lots of other people. *(Cuando salgo de casa, evito permanecer mucho tiempo en sitios cerrados en los que hay muchas personas)* | -.02 | .68 |
| 26. I leave the house without thinking about the time restrictions in the area I live (for example, during the night curfew). *(Salgo de casa sin tener en cuenta las restricciones horarias impuestas donde vivo (por ejemplo, durante el toque de queda nocturno))* | .38 | -.65 |
| 27. When I'm in the street, I sometimes take off my mask. *(Cuando voy por la calle, a veces me quito la mascarilla)* | .44 | -.51 |

external judges assessed the appropriateness of the content of the items, and also their length and wording. They considered the items to be suitable for the aims of this research and the population under study. Of the 27 items, 14 are reverse-worded to control for acquiescence (AC) response bias. This control, in turn, allows (a) the clearest, most univocal factor structure to be obtained from the data [23], and (b) clean content scores not contaminated by AC to be estimated. The item stems can be seen in Table 1. Finally, the online survey did not allow to leave any item without response. More specifically, the items appeared one at a time, and it

was not possible to move on to the next item if the current item was blank. For this reason, there is no missing data in this study.

## Data analysis

The sample was randomly split into two halves. We used the first half as a calibration sample to assess the dimensionality and factor structure of the COCOS questionnaire, and identify the items that functioned poorly. The second half of the sample was used as the validation sample to determine whether the factor solution found in the calibration sample was generalizable. More specifically, in the calibration sample we fitted Exploratory Factor Analysis (EFA) solutions based on the unweighted least squares (ULS) criterion, and determined the most plausible number of factors using the optimal implementation of Parallel Analysis [30]. As several items had extreme distributions, with skewness or kurtosis coefficients higher than 1 in absolute values, the chosen EFA model was nonlinear and based on the polychoric inter-item correlation matrices. Furthermore, we applied the procedure described by Ferrando et al. [31] and Lorenzo-Seva and Ferrando [32] as implemented in the Psychological Test Toolbox [33] program to control for AC bias. This procedure removes the variance due to acquiescent responding from the content items using the information provided by the balanced scale. More specifically, it uses the content-balanced items to identify a factor related to AC, so that the effects of this bias can be removed from both, the inter-item correlations and the individual scores on the content factors. An AC-corrected inter-item correlation matrix was also obtained in the validation sample, and a nonlinear Confirmatory Factor Analysis (CFA) solution based on the previous calibration results was fitted to this matrix by using robust ULS estimation as implemented in the FACTOR program [34]. Both the final EFA solution in the calibration sample and the CFA solution in the validation sample agreed closely, thus suggesting that the results are generalizable to different samples drawn from the target population. Therefore, and so as to use all the information provided by the data, we fitted the CFA solution proposed in the validation sample to the overall sample, by using the same procedure described above. Based on the fitted solution, Bayes Expected-A-Posteriori factor score estimates of the content factor levels were finally obtained for all the 1410 participants, and linearly transformed to T scores (i.e., mean 50 and standard deviation 10) to make interpreting the results easier. In the analyses described below, these scores were used as the content COCOS scores.

The effect of the sociodemographic and other variables on compliance with rules was analysed using $t$-test analyses, analysis of variance (when the Levene test indicated heteroscedasticity, we used the Brown-Forsythe test), post-hoc procedures (the Tuckey or Tamhane test, depending on homoscedasticity), and correlations. We used SPSS 25.0 for these analyses. As these analyses involved a considerable number of comparisons, we only considered the effects with a $p < .01$ to avoid an excessive increase in the experimentalwise error rate.

## Results

### Factor analyses

The first half of the sample was used to determine the most appropriate number of factors to retain and to identify the poorly functioning items. The Kaiser-Meyer-Olkin (KMO) [35] index value was .93, which indicated that the correlation matrix was suitable for factor analysis. The results of Parallel Analysis suggested that only one factor was underlying the data. Therefore, we fitted a unidimensional solution based on the ULS criterion and using the procedure mentioned above to control AC bias. All the items substantially loaded on the single content factor, with loadings ranging from .36 to .77. Furthermore, 10 items loaded non-negligibly on the AC factor, which supports the need to control for this bias. Essential unidimensionality

based on the AC-corrected inter-item correlation was further assessed by using three statistics proposed by Ferrando & Lorenzo-Seva [15], all of which suggested that the data was clearly unidimensional. More specifically, the values of the Unidimensional Congruence (UniCo), the Explained Common Variance (ECV), and the Mean of Items Absolute Value Loadings (MIREAL) were .975, .87 and .20 respectively. Values higher than .95 and .85 for UniCo and ECV, respectively, and lower than .30 for MIREAL suggest that the data can be treated as essentially unidimensional.

On the basis of these results, a unidimensional CFA solution was then fitted to the validation sample data. The goodness of fit of this solution was assessed by using several indices that addressed different facets of fit: (a) Overall fit (Goodness of Fit Index (GFI) = .98, and Root Mean Square of Residuals (RMSR) = 0.055). (b) Comparative fit with respect to the null independence model (Comparative Fit Index (CFI) = .99), and (c) relative fit with respect to model complexity (Root Mean Square Error of Approximation (RMSEA) = .041). GFI and CFI values higher than .90, and RMSEA and RMSR values lower than .07 and .08 respectively are indicative of an acceptable fit [36–38]. So, the results suggest that the unidimensional solution fits the COCOS data quite well in both halves of the sample. Finally, the congruence (Burt-Tucker index) and discrepancy (root mean squared difference) between the loading patterns obtained in the calibration and validation sub-samples were 0.997 and 0.040, respectively. To sum up, the unidimensional solution fitted both sub-samples quite well, and the loading estimates of this solution were virtually the same in both.

As the cross-validation analyses led to virtually the same solution, we fitted the unidimensional CFA solution in the overall sample, and then used the solution obtained to compute content factor score estimates for all the participants as described above. Adequacy and goodness-of-fit results in the total sample were: KMO = .94, GFI = .98, RMSR = .053, CFI = .99, and RMSEA = .047. All these results suggest that the data is suitable for factorising and that the fit of the unidimensional solution is quite good. As for the structural estimates, Table 1 shows the loading values of the items on the content factor, as well as the AC loadings obtained after the inter-item correlation matrix has been corrected. The content pattern exhibits substantial loadings for all of the items. Note, also, that several items have non-negligible loadings on the AC factor. Finally, the marginal reliability of the content factor score estimates was $\rho_{\theta\theta}$ = .92, which can be considered to be quite adequate even for individual assessment purposes.

## Item-level statistics in the COCOS questionnaire

Table 2 shows the means and the standard deviations of the raw COCOS' item scores. Because the FA results clearly show that they all measure a single dimension, the item means can be viewed as indices of item extremeness that show the location of these items on the continuum of compliance that is measured.

The high means on most direct items, above 4.0, and the low means on most reverse-worded items, below 2.0, suggest that they are extreme, so levels of compliance with most pandemic-related restrictions and recommendations are high. However, there is less compliance with the recommendations of not touching own's eyes, nose and mouth (item 4), washing hands when arriving home (item 13), not travelling or going sightseeing (item 14), using the mask face in bars or restaurants when not eating and drinking (item 15), not visiting relatives with whom you do not live (item 16) and not going out to eat with friends (item 19).

Table 2 also shows the minimum score, maximum score, skewness and kurtosis coefficients for each item, as well as the item-rest correlations (i.e. classical item discriminations). All the items had a minimum score of 1 and a maximum score of 5. Furthermore, as can be seen in this table, several items had skewness and kurtosis coefficients higher than 1 in absolute value,

**Table 2. Means, standard deviations, minimum scores, maximum scores, skewness, kurtosis and item-rest correlations in the COCOS questionnaire.**

| Item | Mean | Standard deviation | Minimum | Maximum | Skewness | Kurtosis | Item-rest correlation |
|------|------|-------------------|---------|---------|----------|----------|----------------------|
| 1 | 4.3 | 1.0 | 1 | 5 | -1.5 | 1.9 | .28 |
| 2* | 1.8 | 1.3 | 1 | 5 | 1.3 | 0.3 | .56 |
| 3 | 4.1 | 1.1 | 1 | 5 | -1.2 | 0.9 | .55 |
| 4* | 3.4 | 1.2 | 1 | 5 | -0.4 | -0.7 | .43 |
| 5 | 4.4 | 1.0 | 1 | 5 | -1.8 | 2.8 | .50 |
| 6 | 4.5 | 0.9 | 1 | 5 | -1.9 | 3.8 | .54 |
| 7* | 1.8 | 1.2 | 1 | 5 | 1.3 | 0.5 | .50 |
| 8 | 4.5 | 0.9 | 1 | 5 | -1.9 | 3.7 | .48 |
| 9 | 4.2 | 1.0 | 1 | 5 | -1.2 | 1.3 | .32 |
| 10* | 2.0 | 1.3 | 1 | 5 | 1.1 | -0.1 | .46 |
| 11 | 4.3 | 1.0 | 1 | 5 | -1.4 | 1.5 | .37 |
| 12 | 4.3 | 1.0 | 1 | 5 | -1.5 | 1.9 | .56 |
| 13* | 2.4 | 1.5 | 1 | 5 | 0.5 | -1.2 | .43 |
| 14* | 2.3 | 1.3 | 1 | 5 | 0.7 | -0.8 | .42 |
| 15 | 3.8 | 1.3 | 1 | 5 | -0.8 | -0.6 | .53 |
| 16* | 2.7 | 1.3 | 1 | 5 | 0.2 | -1.2 | .39 |
| 17 | 4.3 | 1.0 | 1 | 5 | -1.7 | 2.3 | .49 |
| 18 | 4.1 | 1.3 | 1 | 5 | -1.2 | 0.2 | .64 |
| 19* | 2.5 | 1.4 | 1 | 5 | 0.4 | -1.2 | .61 |
| 20* | 1.3 | 0.9 | 1 | 5 | 3.1 | 8.9 | .31 |
| 21 | 4.2 | 1.2 | 1 | 5 | -1.3 | 0.7 | .44 |
| 22* | 1.5 | 0.9 | 1 | 5 | 2.2 | 4.4 | .49 |
| 23* | 1.3 | 0.8 | 1 | 5 | 2.9 | 8.5 | .58 |
| 24* | 1.3 | 0.8 | 1 | 5 | 3.0 | 8.7 | .43 |
| 25 | 4.1 | 1.2 | 1 | 5 | -1.2 | 0.4 | .54 |
| 26* | 1.4 | 0.9 | 1 | 5 | 2.6 | 6.2 | .54 |
| 27* | 2.0 | 1.3 | 1 | 5 | 1.0 | -0.2 | .51 |

* Reverse-worded items.

as it was mentioned above. The item-rest correlations ranged from .28 to .64, which, as expected, suggests that they are able to differentiate between participants with high and lower levels of compliance with COVID-19 rules.

## Mean-group differences in the COCOS content scores and the usefulness of preventive measure items

Significant sex differences were found for the COCOS estimated content scores. Table 3 shows the means and standard deviations of these scores both in the whole sample, and by sex (as mentioned above, these scores are scaled in the T metric in the whole group). As can be seen in the table, women showed higher levels of compliance than men, with a medium effect size.

Table 3 also shows the means and standard deviations of the four binary items about how useful particular measures are at preventing the spread of COVID-19. As can be seen, more than 90% of participants considered that social distancing, masks and hand hygiene are useful. However, only about half of the participants considered time and mobility restrictions to be useful. There were significant differences between the opinions of men and women about

**Table 3. Descriptive statistics and sex differences in the COCOS questionnaire and the items on the usefulness of preventive measures.**

|  |  | Mean | S.D. | Men | Women | *p* | *t* | *d* |
|---|---|---|---|---|---|---|---|---|
| COCOS questionnaire |  | 50.0 | 8.78 | 47.6 | 51.7 | < .01 | $t_{(933.3)} = -8.4^*$ | .49 |
| Usefulness of preventive measures | Mask | .96 | .19 | .95 | .97 | >.05 | $t_{(1003.6)} = -1.4^*$ | - |
|  | Social distance | .93 | .25 | .91 | .95 | < .01 | $t_{(911.2)} = -3.1^*$ | .16 |
|  | Hand hygiene | .95 | .21 | .94 | .97 | < .01 | $t_{(902.5)} = -2.6^*$ | .14 |
|  | Time restrictions | .54 | .50 | .59 | .52 | < .01 | $t_{(1170.9)} = 2.6^*$ | .14 |
|  | Mobility restrictions | .59 | .49 | .61 | .58 | >.05 | $t_{(1167.0)} = 1.1^*$ | - |

* Welch's *t*-test.

social distancing, hand hygiene and time restrictions, but the effect sizes were very small, as can be seen in the table.

## Relationships between sociodemographic and attitudinal variables and the COCOS content scores

The validity results in this section aim to provide relational evidence that would support the interpretation of the COCOS content scores as measures of the level of compliance. In principle, these scores are proxies for the construct they measure and, therefore, are attenuated by structural and measurement error. However, given (a) the strong unidimensional structure of the test, and (b) the high reliability of the content scores, the COCOS content scores are regarded as if they were the 'true' scores in the content that is measured. So the results below use the score-external variable correlations (empirical validity coefficients) as if they were the construct-external variable correlations (theoretical validity coefficients).

Age was positively correlated with the level of compliance ($r = .39$, $p < .01$), and according to the criteria proposed by Funder & Ozer [39] the effect size was large. The household income reported by adults was positively correlated with compliance ($r = .11$, p < .01), but the effect size was small. The correlation between compliance and the item on self-perception as an optimist was not significant ($r = .021$, $p >.05$).

Table 4 shows the t-test analyses carried out with several binary variables. As can be seen, both the fear of being infected and the fear of a love one becoming infected are related to higher levels of compliance, with large effect sizes. Likewise, concern about the health crisis is related to higher levels of compliance, also with a large effect size. Both trust in experts and trust in the health system are related to higher levels of compliance, with large and small effect sizes, respectively. Suffering a chronic disease is also related to higher levels of compliance, but the effect size is small. Being vaccinated is not related to compliance, but the intention of being vaccinated in those people who have not been vaccinated does involve higher levels of compliance, with a large effect size. Having had the COVID-19 is not related to compliance. As far as the main sources of information about the pandemic are concerned, the official channels (television, press, government communications, etc.) involve higher levels of compliance, with a medium effect size. The use of social networks (WhatsApp, Facebook, Instagram, etc.) or webs with information other than that offered by the official channels is not related to compliance. However, other sources of information involve lower levels of compliance, with a small effect size. Those participants who considered wearing a mask, social distancing, hand hygiene, time restrictions and mobility restrictions to be useful showed higher levels of compliance, with large effect sizes in the first three cases and medium effect sizes for time and mobility restrictions. Table 4 also shows the means in compliance for adults with and without children. As can be seen, adults with children reported higher levels of compliance, with a small effect size.

**Table 4. Student's t-test to assess the effect of several variables on the level of compliance with the preventive measures.**

| | | Yes | No | t | p | d |
|---|---|---|---|---|---|---|
| Feelings and opinions about the pandemic | Are you afraid of being infected with COVID-19? | 53.3 (n = 687) | 46.9 (n = 723) | $t_{(1180.4)} = 14.8^*$ | < .01 | 0.8 |
| | Are you afraid that a loved one might be infected with COVID-19? | 50.7 (n = 1292) | 42.3 (n = 118) | $t_{(125.6)} = 7.0^*$ | < .01 | 1.0 |
| | Are you worried about the health crisis caused by COVID-19? | 51.6 (n = 1209) | 40.6 (n = 201) | $t_{(228.1)} = 13.3^*$ | < .01 | 1.4 |
| | Do you trust what the experts (epidemiologists, doctors, etc.) have to say about COVID-19? | 51.0 (n = 1214) | 43.9 (n = 196) | $t_{(223.6)} = 8.2^*$ | < .01 | 0.8 |
| | Do you trust the health system? | 50.9 (n = 1056) | 47.2 (n = 354) | $t_{(496.2)} = 6.2^*$ | < .01 | 0.4 |
| Health and vaccination | Chronic disease | 52.2 (n = 216) | 49.6 (n = 1194) | $t_{(312.7)} = 4.3^*$ | < .01 | 0.3 |
| | Vaccinated against COVID-19 | 52.0 (n = 117) | 49.8 (n = 1293) | $t_{(1408)} = 2.6$ | >.01 | - |
| | If not vaccinated, intention to be vaccinated against COVID-19 | 51.1 (n = 1083) | 43.1 (n = 210) | $t_{(249.4)} = 10.1^*$ | < .01 | 0.8 |
| | Being infected with COVID-19 at some point | 49.1 (n = 131) | 50.1 (n = 1279) | $t_{(1408)} = -1.3$ | >.05 | - |
| | If infected with COVID-19 at some point, having suffered severe symptoms | 52.1 (n = 20) | 48.5 (n = 111) | $t_{(129)} = 1.6$ | >.05 | - |
| Main source of information | Official channels | 50.7 (n = 1229) | 45.3 (n = 181) | $t_{(206.3)} = 6.0^*$ | < .01 | 0.6 |
| | Social networks | 49.5 (n = 631) | 50.4 (n = 779) | $t_{(1408)} = -2.1$ | >.01 | - |
| | Webs with alternative information | 49.8 (n = 279) | 50.0 (n = 1131) | $t_{(1408)} = -0.3$ | >.05 | - |
| | Others | 48.1 (n = 188) | 50.3 (n = 1222) | $t_{(221.6)} = -2.7^*$ | < .01 | 0.3 |
| Opinion about the usefulness of the preventive measures | Usefulness of mask | 50.5 (n = 1355) | 37.9 (n = 55) | $t_{(56.0)} = 7.6^*$ | < .01 | 1.5 |
| | Usefulness of social distancing | 50.9 (n = 1318) | 37.3 (n = 92) | $t_{(97.1)} = 11.2^*$ | < .01 | 1.7 |
| | Usefulness of hand hygiene | 50.5 (n = 1345) | 39.7 (n = 65) | $t_{(66.9)} = 7.1^*$ | < .01 | 1.3 |
| | Usefulness of time restrictions | 52.3 (n = 768) | 47.3 (n = 642) | $t_{(1139.2)} = 10.8^*$ | < .01 | 0.6 |
| | Usefulness of travel restrictions | 51.7 (n = 833) | 47.6 (n = 577) | $t_{(1008.5)} = 8.4^*$ | < .01 | 0.5 |
| | Having children | 54.2 | 51.1 | $t_{(924.1)} = 6.8$ | < .01 | 0.4 |

$^*$ Welch's t-test.

Table 5 shows the ANOVA results between the civil status of adults and the level of compliance. Significant differences were found for the civil status of adults, with a low effect size. Married people obtained the highest mean in compliance, but pairwise mean comparisons only showed significant differences with singles (difference = 3.7, $p < .01$, $d = 0.56$), and with people living with a partner (but not married) (difference = 3.9, $p < .01$, $d = 0.55$). There were no significant differences with other civil status groups. Furthermore, the ANOVA results between the educational level in adults and compliance were not significant ($F(4, 1012) = 0.58$, $p = .68$).

**Table 5. Analysis of variance of the effect of civil status on compliance with the preventive measures.**

| | | Mean | S.D. | F | p | η² |
|---|---|---|---|---|---|---|
| Civil status of adults | Single | 51.0 | 7.4 | | | |
| | Married | 54.7 | 5.7 | | | |
| | With a partner (not married) | 50.8 | 8.2 | 10.2* | < .01 | .06 |
| | Divorced/separated | 53.2 | 6.6 | | | |
| | Widowed | 51.6 | 12.9 | | | |

*The Brown-Forsythe Statistic was used due to lack of homoscedasticity.

## Discussion

The main goal of the current study was to develop a new questionnaire to assess the level of compliance with COVID-19 preventive measures which (a) included a more comprehensive set of measures than those in previous studies, (b) controlled the impact of acquiescence bias, and (c) had appropriate psychometric properties, including dimensionality, structure, accuracy of the derived scores, and meaningful relations with relevant external variables. Some previous studies obtained multiple correlated-factor solutions, usually associated with weak reliabilities of the derived scores [e.g., 14,16]. However, our results based on the COCOS data suggested that only one content factor was underlying the data. Our proposed solution was strong, stable across different subsamples, and allowed reliable scores to be derived from it. Furthermore, several items had substantial loadings on the AC factor, which shows the relevance of controlling for this response bias in this kind of instrument.

The relationships found between the COCOS scores and the sociodemographic variables are similar to those obtained in previous studies, which is evidence of the validity of the COCOS questionnaire. In fact, previous studies had already shown higher levels of compliance in women [1,2,4], older people [4,6], married people [2], and adults with children [2], as we also found here. Some previous results suggest that the level of income does not play a relevant role in the prediction of compliance [2,8]. In the current study we found a significant correlation between these variables, but with a small effect size, which also suggests that income is not highly related to compliance.

Our results on attitudes and opinions about the pandemic are also similar to those found in other studies. More specifically, and as expected, people who trust in experts and in the health system had higher levels of compliance. Several studies in different countries also found that people who trust the health system [9] and the government [1,2] are more likely to comply with the measures. Also as expected, people who considered masks, social distancing, hand hygiene, time restrictions and mobility restrictions to be useful tended to report higher levels of compliance. Furthermore, people who were afraid of getting infected, afraid of a loved one getting infected, or worried about the health crisis tended to display higher levels of compliance. Jørgensen et al. [40] and Šuriņa et al. [41] also found that fear and perceptions of threat were related to greater compliance.

According to Monzani et al. [12], optimistic people may develop an optimistic bias about the COVID-19, judging that they have a lower risk of infection than others. This could lead to less engagement with preventive measures. We did not find a relationship between optimism and compliance here. This result, however, may be explained by the fact that we assessed optimism with only one item which referred solely to self-perception of optimism rather than optimistic bias.

People who tend to use official channels to get information reported higher levels of compliance. The study by Wang et al. [11] showed that receiving mixed messages about the pandemic may lead to lower levels of compliance. Since official channels such as the press, television, government communications, etc., tend to provide more updated and reliable information, and less mixed information than other channels, they may lead to higher compliance, as the current study suggests. Furthermore, the study by Morales-Vives et al. [42] during the national lockdown in Spain in 2020 showed that people who sought information about the pandemic only from official media tended to adapt to the lockdown situation better, with less stress and negative emotions, and more positive attitudes and behaviors. Therefore, these studies show the positive impact of the official channels during the different stages of the pandemic.

We also aimed to determine whether being vaccinated (having received at least one dose of a vaccine) leads to lower levels of compliance because the vaccine may provide a certain sense

of security. However, we found no relation, which suggests that vaccinated people remain vigilant and do not ignore the preventive measures. However, further studies should be done on this issue, particularly when more sectors of the population, especially the young, get vaccinated. Another positive outcome is the high levels of compliance reported by the participants for most measures. The lowest measures of compliance were for not touching one's own eyes, nose and mouth, washing hands when arriving home, not travelling or going sightseeing, using the face mask in bars or restaurants when not eating or drinking, not visiting relatives with whom you do not live and not eating out with friends. Two of these measures refer to bars and restaurants (eating out with friends, and not using the mask in bars and restaurants), which suggests that they are playing an important role in spreading the infection in Spain.

Regarding the limitations of this study, it should be taken into account than 57% of the sample were adolescents or young people under 30 years old, possibly because younger people tend to use new technologies to a greater extent, making it easier for them to participate in online studies. Therefore, the age distribution of this sample does not fully correspond to the age distribution in the Spanish population, and for this reason the age-compliance relationship should be interpreted cautiously. However, the number of adults in this study cannot be considered as totally insufficient or inadequate, as 43% of the participants were aged 30 years or older.

In summary, the current study first shows that the COCOS questionnaire has adequate internal psychometric properties. As for validity evidence, the relationships between compliance and the other variables are similar to those provided in previous studies. Taking into account that ours is a more comprehensive instrument, we submit it is a useful measure for assessing the level of compliance in the target population. The results of the current study also show the importance of explaining the risks of infection, and the benefits of preventive measures, and of promoting trust in the government, the health system and the official channels, which provide reliable information about the pandemic. These explanations should be especially addressed to people with lower compliance, such as young people or men.

Further COCOS-based studies are needed in other samples to collect more evidence about the replicability of the present results and the validity of the COCOS scores.

We would like to finish by noting that many of the previous instruments were developed to assess compliance during lockdown periods, while the current study was done during the so-called new normality (the disease is still present, but the measures are not as restrictive as a general lockdown). It is important to get evidence about the behaviors and opinions of the population during this period because preventive measures are still necessary to prevent the pandemic from spreading and to protect the most vulnerable people. Finally, some of the preventive measures discussed in this questionnaire may no longer be necessary at future stages of the pandemic. In this case, the COCOS could be tailored to include only the measures deemed to be relevant at the time, while still maintaining its good psychometric properties. Its simplicity, strong, positive-manifold structure, and the reliability of the scores warrant shorter versions with only marginal losses in accuracy. Furthermore, a short version of this questionnaire should be developed to be used in clinical and practical settings. In fact, several studies show that short questionnaires are useful in this kind of settings because they save time and do not tire the participants [43–45].

## Author Contributions

**Conceptualization:** Fabia Morales-Vives, Andreu Vigil-Colet.

**Data curation:** Fabia Morales-Vives, Andreu Vigil-Colet.

**Formal analysis:** Fabia Morales-Vives, Pere J. Ferrando, Andreu Vigil-Colet.

**Funding acquisition:** Fabia Morales-Vives.

**Investigation:** Fabia Morales-Vives, Jorge-Manuel Dueñas.

**Methodology:** Fabia Morales-Vives, Pere J. Ferrando, Andreu Vigil-Colet.

**Project administration:** Fabia Morales-Vives, Jorge-Manuel Dueñas, Andreu Vigil-Colet, Maria Dolores Varea.

**Resources:** Jorge-Manuel Dueñas, Maria Dolores Varea.

**Software:** Jorge-Manuel Dueñas.

**Supervision:** Fabia Morales-Vives, Andreu Vigil-Colet.

**Validation:** Fabia Morales-Vives, Pere J. Ferrando.

**Visualization:** Fabia Morales-Vives, Jorge-Manuel Dueñas.

**Writing – original draft:** Fabia Morales-Vives, Jorge-Manuel Dueñas.

**Writing – review & editing:** Fabia Morales-Vives, Pere J. Ferrando, Andreu Vigil-Colet.

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
