## [Decision Letter · Decision Letter 0]

22 Dec 2021

PONE-D-21-23556COmpliance with pandemic COmmands Scale (COCOS): The relationship between compliance with COVID-19 measures and sociodemographic and attitudinal variablesPLOS ONE

Dear Dra. Fabia Morales,

Thank you for submitting your manuscript to PLOS ONE. After careful consideration, we feel that it has merit but does not fully meet PLOS ONE’s publication criteria as it currently stands. Therefore, we invite you to submit a revised version of the manuscript that addresses the points raised during the review process.

Please submit your revised manuscript by21/02/2022. If you will need more time than this to complete your revisions, please reply to this message or contact the journal office at plosone@plos.org. Please include the following items when submitting your revised manuscript:A rebuttal letter that responds to each point raised by the academic editor and reviewer(s). You should upload this letter as a separate file labeled 'Response to Reviewers'.A marked-up copy of your manuscript that highlights changes made to the original version. You should upload this as a separate file labeled 'Revised Manuscript with Track Changes'.An unmarked version of your revised paper without tracked changes. You should upload this as a separate file labeled 'Manuscript'.If applicable, we recommend that you deposit your laboratory protocols in protocols.io to enhance the reproducibility of your results. Protocols.io assigns your protocol its own identifier (DOI) so that it can be cited independently in the future. For instructions see: https://journals.plos.org/plosone/s/submission-guidelines#loc-laboratory-protocols. Additionally, PLOS ONE offers an option for publishing peer-reviewed Lab Protocol articles, which describe protocols hosted on protocols.io. Read more information on sharing protocols at https://plos.org/protocols?utm_medium=editorial-email&utm_source=authorletters&utm_campaign=protocols.

We look forward to receiving your revised manuscript.

Kind regards,

Eduardo Fonseca-Pedrero, PhD

Academic Editor

PLOS ONE

Journal Requirements:

2. PLOS ONE does not copy edit accepted manuscripts (https://journals.plos.org/plosone/s/criteria-for-publication#loc-5). To that effect, please ensure that your submission is free of typos and grammatical errors.

*Peer review at PLOS ONE is not double-blinded (https://journals.plos.org/plosone/s/editorial-and-peer-review-process). For this reason, authors should include in the revised manuscript all the information removed for blind review.

“This research was supported by a grant from the Spanish Ministry of Economy and Competitivity (PID2020-112894GB-I00) and a grant from the Catalan Ministry of Universities, Research and the Information Society (2017 SGR 97).”

Funded study:

- Initials of authors who received any grant: FMV, JMD, PJFP and AVC

- Grant numbers awarded to FMV, JMD, PJFP and AVC: PID2020-112894GB-I00 and 2017 SGR 97

- Full name of funder: Spanish Ministry of Economy and Competitivity (PID2020-112894GB-I00)

- Full name of funder: Catalan Ministry of Universities, Research and the Information

Society (2017 SGR 97).

- URL of the Spanish Ministry of Economy and Competitivity:

https://portal.mineco.gob.es/es-es/Paginas/default.aspx

- URL of the Catalan Ministry of Universities, Research and the Information Society:

https://agaur.gencat.cat/ca/inici

The funders had no role in study design, data collection and analysis, decision to

publish, or preparation of the manuscript”

Reviewers' comments:

Reviewer's Responses to Questions

**Comments to the Author**

1. Is the manuscript technically sound, and do the data support the conclusions?

Reviewer #1: Yes

Reviewer #2: Yes

2. Has the statistical analysis been performed appropriately and rigorously? 

Reviewer #1: Yes

Reviewer #2: Yes

3. Have the authors made all data underlying the findings in their manuscript fully available?

Reviewer #1: Yes

Reviewer #2: No

4. Is the manuscript presented in an intelligible fashion and written in standard English?

Reviewer #1: Yes

Reviewer #2: Yes

5. Review Comments to the Author

Reviewer #1: El manuscrito COmpliance with pandemic COmmands Scale (COCOS): The relationship between compliance with COVID-19 measures and sociodemographic and attitudinal variables es una pieza de investigación atractiva e interesante: los autores justifican adecuadamente la pertinencia del estudio y, en general, tiene nivel suficiente para publicarse en PLOS-ONE. No obstante, advierto algunos aspectos mejorables y, por tanto, me resisto a recomendar su publicación hasta que sean resueltos.

- La tabla 1 y 2 tienen información redundante. En ambas aparece las Medias y SD por ítem (además las medias no coinciden en el caso de 4 ítems: revisar). Recomiendo quitar los descriptivos de la tabla 1 y, puesto que la tabla 2 está en el apartado “estadísticas del ítem” completar la misma con otra información de los ítems. Por ejemplo, en Data Analysis los autores justifican el uso de un modelo no-lineal por la asimetría y curtosis de varios ítems. Pueden, por tanto, en la tabla 2 dar esos valores de los ítems. También podría ser interesante conocer el rango de los ítems (o los mínimos-máximos) y otros valores como la correlación ítem-tests o la aportación de cada ítem a la consistencia interna de la escala. En definitiva, si un apartado es “estadísticas de los ítems” informen de las mismas de forma exhaustiva y, en todo caso, eviten reiteraciones de información.

- En la página 17 también me parece reiterativo las dos llamadas seguidas a la tabla 3. Dejen un apartado para las estadísticas de los ítems y otro para las comparaciones de medias. En todo caso, ¿por qué las comparaciones de género sólo se hacen sobre el uso de las medidas preventivas y no sobre el resto de variables que emplean al margen de la escala?

- El trabajo parece tener dos partes bien diferenciadas. La primera es la validación de la escala CoCos y la segunda cruzar los resultados de la escala con algunas variables relevantes. La primera parte está muy bien ejecutada: bien descrito los procedimientos de análisis, buenos índices de ajuste, obteniendo una solución unidimensional y consistente.

- Sin embargo, la segunda parte, por momentos, me ha resultado algo confusa obligándome a ir de los Resultados al Método para confirmar algunas cuestiones. Por ejemplo, en la Tabla 4 aparecen cinco ítems (desde Cronic disease hasta If infected…) de los que no se informan en el apartado metodología, restando coherencia al manuscrito.

- El hecho de que las medias de algunos ítems no coincidan en las tablas 1 y 2 me ha hecho pensar si las pequeñas variaciones se deben al modo de tratar los datos perdidos los análisis de cada una de las tablas. En realidad, los autores no dicen nada del tratamiento de los datos perdidos: ¿se configuró la encuesta para que no hubiera datos perdidos?, y si no fue así ¿entre qué porcentajes de datos perdidos oscilan los ítems y cómo se recuperaron los mismos? Me gustaría ver un par de líneas sobre ese aspecto.

- La muestra es bastante grande. Sin embargo, el procedimiento de muestreo y sobre todo el hecho de que se tratara de una aplicación exclusivamente on-line, supone una limitación y un sesgo de autoselección. Creo que eso debería comentarse entre las limitaciones del trabajo: es probable que la distribución por grupos de edad no recupere adecuadamente los porcentajes de grupos de edad en España. ¿puede ser que en su estudio los grupos de edad más jóvenes (o medina edad) estén sobre- representados, ya que acceden de las nuevas tecnologías con mayor frecuencia que las personas con más edad. Eso, por ejemplo, podría ser una amenaza para sus conclusiones sobre la relación Edad-Resultado en COCOS.

En definitiva, mi valoración de manuscrito es satisfactoria y mis recomendaciones son más metodológicas y de estilo, antes que de contenido fundamental. Espero que los autores puedan dar respuesta a todas ellas para ver publicado el manuscrito en Plos-One.

Reviewer #2: En primer lugar, agradecer a la revista y al editor la posibilidad de revisar este manuscrito. El presente estudio muestra un tema interesante y actual que está afectando a la vida de todas las personas, como es el cumplimiento de las medidas en torno al COVID-19. La metodología del estudio es adecuada y los hallazgos son muy interesantes y están bien documentados. No obstante, propongo algunas sugerencias que obtuve al leer el manuscrito:

- La introducción es concisa y clara. No obstante, creo que habría que mirar los objetivos detalladamente, siendo mejor ordenados, ya que el orden no es el mismo que se sigue en los resultados.

- Referido al ítem de optimismo, por la descripción parece que se refiere más a que una persona sea optimista de manera general y no respecto al COVID-19. Clarificar si es así.

- Procedimiento, página 10. En las líneas 227-250. Referido a las recomendaciones en España y a nivel internacional, ¿cuáles fueron esas medidas? ¿Se siguió algún checklist que llevara a que desarrollar ítems sobre diferentes facetas (v.g. contacto social, tema relacionado con mascarilla, higiene…)? Si es así, ¿tendría sentido plantear otro modelo (v.g. bifactor) que contemple facetas aparte de la dimensión general? Por ejemplo, en la discusión (líneas 446-448) se habla de “masks, social distancing, hand higiene, time restrictions and mobility restrictions”. Solo es una duda que se me plantea, aunque sí veo relevante incluir esta información más profundamente en el manuscrito, que lleve a entender a partir de qué recomendaciones se llegó a un banco de 27 ítems.

- La Tabla 1 y Tabla 2 repiten la media y la desviación típica de cada ítem, con una vez es suficiente. Sumado a ello, creo que sería pertinente añadir algún estadístico descriptivo más de cada ítem (v.g. asimetría y curtosis), así como la capacidad discriminativa (v.g. correlación ítem-test corregida).

- Probablemente se me haya escapado, pero no encuentro la fiabilidad en el manuscrito. Esto es importante a mi juicio, y más cuando en la introducción se critica la fiabilidad de otras escalas de la misma índole.

- En línea con el primer comentario, creo que la sección de resultados es un poco liosa a partir de la página 17. ¿A qué objetivo responde respecto a la validación de la escala la Tabla 3? Si después en la página 20 se van a mostrar las diferencias en la escala COCOS en función de variables sociodemográficas, ¿no tendría más sentido añadir aquí las diferencias de sexo que se muestran en la Tabla 3? Sumado a ello, y a pesar de que sea muy interesante, las diferencias de sexo en función de las medidas preventivas que se muestran en la tabla 3 no les veo relación con la escala y provoca confusión. Creo que se debería de poner al final como un objetivo diferente a la validación de la escala o dejarlo para otro estudio diferente.

- Referido a la Tabla 4, ¿sería posible aportar la n de cada grupo? Creo que sería muy clarificador para el lector.

- Referido al optimismo, ¿dónde se encuentra esta variable en los resultados?

- La discusión está correctamente estructurada. Como sugerencia, viendo que se trata de una escala que se puede aplicar en multitud de contextos práctico-clínicos, añadiría como línea futura generar una versión corta de la escala, siendo de gran utilidad por el ahorro de tiempo que supone en estos contextos (e.g. Blanca et al., 2020; García-Alba et al., 2021; Postigo et al., 2020).

Blanca, M. J., Escobar, M., Lima, J. F., Byrne, D., & Alarcon, R. (2020). Psychometric properties of a short form of the Adolescent Stress Questionnaire (ASQ-14). Psicothema, 32(2), 261-267. https://doi.org/10.7334/psicothema2019.288

García-Alba, L., Postigo, Á., Gullo, F., Muñiz, J., & Del Valle, J. F. (2021). PLANEA Independent Life Skills Scale: Development and Validation. Psicothema, 33(2), 268-278. https://doi.org/10.7334/psicothema2020.450

Postigo, Á., García-Cueto, E., Cuesta, M., Menéndez-Aller, Á., Prieto-Díez, F., y Lozano, L. M. (2020). Assessment of the enterprising personality: A short form of the BEPE battery. Psicothema, 32(4), 575–582. https://doi.org/10.7334/psicothema2020.193

Aspectos menores: El ítem 8 no se ve adecuadamente en la tabla 1.

6. PLOS authors have the option to publish the peer review history of their article (what does this mean?). If published, this will include your full peer review and any attached files.

Reviewer #1: No

Reviewer #2: **Yes: **Álvaro Postigo

---

## [Author Response · Author response to Decision Letter 0]

30 Dec 2021

Response to the Editor:

1. As the editor suggests, we have reviewed the PLOS ONE's style requirements, and we have carried out several corrections in the manuscript and the title page. Furthermore, we have changed the name of the file.

2. We have reviewed the manuscript and we have made some corrections, to avoid typos and grammatical errors. Furthermore, we have included in the revised manuscript all the information previously removed for blind review.

3. As funding information should not appear in the Acknowledgments section or other areas of the manuscript, we have removed the Acknowledgements section of our manuscript. As suggested by the Editor, we have included our amended statement within our cover letter, in order you change the online submission form on our behalf.

4. We have reviewed our reference list to ensure that is correct and complete. Furthermore, we have introduced the three references suggested by Reviewer 2. On the other hand, we have not cited any paper that have been retracted.

Response to Reviewer 1:

Le agradecemos al revisor/a sus sugerencias en relación al manuscrito, ya que han sido muy útiles para mejorar el artículo. De hecho, a partir de estos comentarios, hemos introducido diversos cambios en el texto. A continuación explicamos estos cambios, y respondemos las diferentes cuestiones que ha planteado.

1. Tal como el revisor ha señalado, la tabla 1 y 2 proporcionaban información redundante, ya que ambas mostraban las medias y desviaciones típicas de los ítems. Por ese motivo, hemos eliminado estos valores de la Tabla 1. Las diferencias en las medias de cuatro de los ítems se debían al redondeo de los decimales. Hemos revisado todos los valores, de tal forma que ahora el redondeo sigue el siguiente patrón: cuando el segundo decimal es igual o inferior a 4, hemos redondeado a la baja, y cuando el segundo decimal es de 5 o más, hemos redondeado al alza. Por otra parte, hemos teniendo en cuenta las sugerencias del revisor y hemos añadido más información en la Tabla 2. Concretamente, esta tabla muestra ahora los valores mínimos y máximos de cada ítem y los coeficientes de asimetría y curtosis.

2. Tal como sugiere el revisor, hemos dejado un apartado exclusivamente para las estadísticas de los ítems y otro apartado para las comparaciones de medias.

Dado que el estudio está centrado en el seguimiento de las medidas preventivas, tan sólo hemos evaluado las diferencias entre hombres y mujeres en el seguimiento de estas medidas y en la opinión sobre la utilidad de las mismas. Estamos de acuerdo en que sería interesante proporcionar también resultados sobre las diferencias de sexo en las otras variables, pero hemos considerado que estos análisis alargarían mucho un artículo que ya es de por sí relativamente largo, cuando estos análisis van más allá de los objetivos planteados inicialmente en el estudio, ya que el resto de variables no se refieren directamente a las medidas preventivas.

3. Le agradecemos al revisor sus comentarios positivos en relación a los procedimientos y resultados de la primera parte del estudio. En relación a la segunda parte, estamos de acuerdo en que la tabla 4 podía resultar confusa, en parte porque algunos de los ítems no estaban incluidos en el apartado Measures, tal como ha señalado el revisor. Por ese motivo, hemos añadido un nuevo párrafo en el apartado Measures, con los ítems que faltaban. También hemos hecho diversos cambios en la Tabla 4 para facilitar su interpretación, y para que resulte más fácil enlazarla con el apartado Measures. Le agradecemos la sugerencia al revisor, dado que consideramos que este apartado resulta ahora más claro que antes.

4. La aplicación que usamos para completar los cuestionarios no permitía dejar ningún ítem en blanco. De hecho, los ítems aparecían de uno en uno, y no se podía pasar al ítem siguiente si el actual estaba en blanco. Por ese motivo, no hay datos perdidos. Por lo tanto, las diferencias en las medias de cuatro ítems en la tabla 1 y 2 no se deben a la existencia de datos perdidos, si no al redondeo de decimales, como hemos explicado anteriormente. Hemos especificado en el artículo que no hay datos perdidos.

5. Tal como el revisor ha planteado, los adolescentes y jóvenes están bien representados en la muestra, probablemente porque son lo que más utilizan las nuevas tecnologías, y por lo tanto son los que más participan en estudios que se realizan de forma online. Pero eso no implica que haya muy pocos adultos con más de 30 años en este estudio, ni que el número de adultos de estas edades sea totalmente insuficiente o inadecuado. Concretamente, en la muestra hay un 28% de adolescentes, un 29% de jóvenes entre los 18 y 29 años, y un 43% de adultos con 30 años o más. Por otra parte, el 31% de la muestra tiene 40 años o más. De hecho, realizamos una ardua tarea de difusión para lograr que la muestra fuera lo más heterogénea posible, y muestra de ello es que en este estudio incluso han participado personas con más de 60 años (70 personas). Sin embargo, a pesar de toda la difusión realizada, el 57% de la muestra tiene menos de 30 años, por lo que hemos reconocido esta limitación en la Discusión.

Response to Reviewer 2:

Le agradecemos al revisor sus comentarios y sugerencias, dado que nos han ayudado a clarificar y mejorar algunos aspectos del artículo. A continuación exponemos los cambios que hemos realizado en el manuscrito y respondemos a las cuestiones planteadas por el revisor.

1. Hemos ordenado mejor los objetivos, de tal forma que ahora parecen en el mismo orden que en los resultados.

2. Tal como dice el revisor, el contenido del ítem de optimismo es general, no se refiere específicamente a la COVID-19. Concretamente, el ítem es el siguiente: “En una escala de 0 a 10, ¿hasta qué punto te consideras una persona optimista?”. Hemos especificado en el texto que el ítem no se refiere a la COVID-19, si no al optimismo en general.

3. Como recomienda el revisor, hemos explicado con más detalle qué medios y entidades se consultaron para obtener información sobre las medidas preventivas recomendadas, incluyendo las citas pertinentes. Concretamente, consultamos las recomendaciones propuestas por la OMS y por el Centers for Disease Control and Prevention, las medidas recomendadas por el Ministerio de Sanidad, Consumo y Bienestar Social y por las comunidades autónomas, y también las medidas incluidas en investigaciones previas. En general, las medidas se engloban en los diferentes ámbitos: uso de la mascarilla, distancia social, higiene de manos, cubrir la nariz y la boca con el codo flexionado o con un pañuelo, cumplimiento de los toques de queda y de las restricciones de movilidad impuestas por las diferentes administraciones, aislamiento en caso de presentar síntomas o de ser diagnosticado de COVID-19, aislamiento en caso de ser contacto de una persona diagnosticada de COVID-19, y otros (evitar fiestas, aglomeraciones, transporte público, dar la mano, viajar sin una razón justificada, compartir bebida y comida con personas con las que no convives, o tocarse los ojos, nariz y boca). Por este motivo, se redactaron ítems de cada una de estas áreas, como puede observarse en la Tabla 1.

Considerando que se partía de diferentes facetas, el revisor pregunta si hubiera tenido sentido plantear algún otro modelo que contemplara estas facetas, como por ejemplo un bifactor. La idea, desde luego, tiene sentido. Sin embargo, los resultados sugieren que estos datos tienen un ajuste excelente al modelo undimensional y, por tanto, que, al extraer el factor común, los residuales virtualmente desaparecen. El modelo bifactor es útil cuando, además de un factor dominante, existen factores de grupo justificables que impiden un buen ajuste del modelo unidimensional. Sin embargo, tratar de extraer factores de grupo a partir de residuales que ya son prácticamente nulos, es un caso de sobre-factorización y no es justificable. Para ver este punto desde una perspectiva más específica, cabe notar que el valor del índice MIREAL es inferior a 0.30. Este índice indica cuál es el promedio de las saturaciones secundarias en sucesivos factores una vez se ha extraído el factor principal. Por tanto, si se intentara extraer adicionalmente un solo factor de grupo (no digamos más de uno) este factor no estaría ni siquiera mínimamente determinado. Por ese motivo, no optamos por la opción de un bifactor. Los resultados sugieren que hay un único factor subyacente a los diferentes ítems y facetas, y por ese motivo no hemos probado otros modelos.

4. Tal como señala el revisor, la tabla 1 y 2 eran redundantes, porque ambas mostraban las medias y desviaciones típicas de los ítems. Por ese motivo, hemos eliminado esta información de la tabla 1. Además, tal como sugiere el revisor, hemos añadido información adicional en la tabla 2: coeficientes de asimetría y curtosis, valores mínimo y máximo en cada ítem, y correlaciones ítem-total corregidas.

5. La fiabilidad se muestra en el texto, concretamente en las líneas 392-394 de la página 15. La frase es la siguiente: Finally, the marginal reliability of the content factor score estimates was ��� = .92, which can be considered to be quite adequate even for individual assessment purposes”.

6. y 7. Hemos reformulado el párrafo de los objetivos, de tal forma que la parte de la investigación referida a las diferencias de sexo (Tabla 3) está ahora mejor justificada, y más enlazada con el resto del estudio. Por lo tanto, los análisis de la Tabla 3 están ahora bien incluidos en los objetivos, de tal forma que se justifica mejor que se presenten en una tabla propia. Es cierto que esta parte no constituye el objetivo principal ni central del estudio, pero consideramos que era interesante añadirla, dado que estudios previos ya habían mostrado diferencias de sexo en el cumplimiento de las medidas preventivas. Por lo tanto, uno de nuestros objetivos era determinar si estas diferencias también se observaban con el cuestionario COCOS, en población española.

8. Tal como ha solicitado el revisor, hemos aportado la n para cada grupo en la Tabla 4.

9. El revisor pregunta donde se encuentra la variable optimismo en los resultados. Concretamente, la correlación entre optimismo y cumplimiento de las medidas preventivas está en la página 21 del documento, líneas 483-484. La frase es la siguiente: “The correlation between compliance and the item on self-perception as an optimist was not significant (r = .021, p >.05).” 

10. Tal como sugiere el revisor, hemos añadido en la discusión que en el futuro se podría desarrollar una versión corta del cuestionario, ya que sería útil en contextos práctico-clínicos. Hemos utilizado las referencias que el revisor había puesto como ejemplo, para justificar la utilidad de una versión más corta del cuestionario.

11. Tal como el revisor ha señalado en relación a la tabla 1, el ítem 8 no se veía bien, ya que la fila no era lo suficientemente ancha como para mostrar todo el texto. Por ese motivo, hemos ampliado esa fila, de tal forma que ahora se puede leer todo el texto.

---

## [Editor Report · Decision Letter 1]

4 Jan 2022

COmpliance with pandemic COmmands Scale (COCOS): The relationship between compliance with COVID-19 measures and sociodemographic and attitudinal variables

PONE-D-21-23556R1

Dear Dr. Fabia Morales,

We’re pleased to inform you that your manuscript has been judged scientifically suitable for publication and will be formally accepted for publication once it meets all outstanding technical requirements.

Kind regards,

Eduardo Fonseca-Pedrero, PhD

Academic Editor

PLOS ONE
---

## [Editor Report · Acceptance letter]

10 Jan 2022

PONE-D-21-23556R1 

COmpliance with pandemic COmmands Scale (COCOS): The relationship between compliance with COVID-19 measures and sociodemographic and attitudinal variables 

Dear Dr. Morales-Vives:

I'm pleased to inform you that your manuscript has been deemed suitable for publication in PLOS ONE. Congratulations! Your manuscript is now with our production department. 

Kind regards, 

on behalf of

Dr. Eduardo Fonseca-Pedrero 

Academic Editor

PLOS ONE